# Investigating Owner Use of Dietary Supplements in Dogs with Canine Cognitive Dysfunction

**DOI:** 10.3390/ani13193056

**Published:** 2023-09-29

**Authors:** Julia Haake, Nina Meyerhoff, Sebastian Meller, Friederike Twele, Marios Charalambous, Volker Wilke, Holger Volk

**Affiliations:** 1Department of Small Animal Medicine and Surgery, University of Veterinary Medicine Hannover, 30559 Hannover, Germany; julia.patricia.haake@tiho-hannover.de (J.H.); nina.meyerhoff@tiho-hannover.de (N.M.); sebastian.meller@tiho-hannover.de (S.M.); friederike.twele@tiho-hannover.de (F.T.); marios.charalambous@tiho-hannover.de (M.C.); 2Institute for Animal Nutrition, University of Veterinary Medicine, Foundation, 30173 Hannover, Germany; volker.wilke@tiho-hannover.de

**Keywords:** veterinary neurology, canine cognitive dysfunction, dietary supplements, diet, geriatrics

## Abstract

**Simple Summary:**

Canine cognitive dysfunction is considered the canine equivalent to human Alzheimer’s disease. It is a growing concern in veterinary medicine, as it affects many aged dogs. Dietary intervention with different diets and supplements may improve clinical signs and prevent further degeneration. Using an online questionnaire, we found that even though few owners were willing to change their dog’s main diet, many of them added supplements such as oils and vitamins. Consulting a veterinary surgeon when using dietary supplements is important as it allows for evidence-based recommendations to be made.

**Abstract:**

Canine cognitive dysfunction (CCD) is becoming increasingly recognized in veterinary medicine, as dogs live longer and with CCD being highly prevalent among the elderly dog population. Various studies have shown that diet and dietary supplementation can positively influence the clinical signs of CCD, especially if given at an early stage. The aim of this study was to investigate owner use of dietary supplements (DSs) in dogs with age-related behavioral changes. An observational study based on an online questionnaire for owners of dogs with age-related behavioral changes was performed. Out of a total of 394 owners who completed the survey, after noticing age-related behavioral changes, over half of the dogs received DSs (54%), whereas only 8% reported changing their dog’s base diet. The most used DS was fish oil (48%). The use of DSs should be discussed with and monitored by veterinary surgeons since many geriatric patients have multi-morbidities, may have specific nutritional requirements and receive multi-faceted medications.

## 1. Introduction

Age-related degenerative illnesses such as canine cognitive dysfunction (CCD) are becoming more relevant for veterinary medicine, as a result of pets’ longer life expectancies [1]. Various studies describe the prevalence ranging from 14.2% to 68% in geriatric dogs [2,3,4]. As these patients can exhibit various clinical signs and behavioral changes such as disorientation, confusion, incontinence, altered activity, and changes in sleep–wake rhythms [5,6], the quality of life can be severely affected in dogs and their caretakers [7].

CCD is considered to be similar to Alzheimer’s disease (AD) in humans, as comparable neuropathological and metabolic changes can be found in the brains of both affected humans and dogs [5,8]. ß-amyloid plaques accumulate in the brains of both dogs with CCD and humans with AD. These peptides can be found in the brain parenchyma and cerebral blood vessels [9]. Therefore, they can cause cerebral amyloid angiopathy, which may lead to reduced blood flow, infarctions, microhemorrhages, and hypoxia [10,11,12,13]. Neurofibrillary tangles (NFTs), which are commonly found in the brains of AD patients, are not present in dogs with CCD [14], but accumulations of hyperphosphorylated protein (tau protein), a precursor to NFTs, can be found [15]. Further neuropathological findings include cerebral atrophy and ventricular dilation due to neuronal loss, which may be caused by decreased neurogenesis, oxidative stress, and chronic inflammation [16,17].

Physiologically, the neurons of the brain require a large amount of energy, which is mostly derived from glucose metabolism [18], in which glucose is converted to ATP by the mitochondria. In the absence of glucose, e.g., in periods of starvation, the brain relies on alternative energy sources such as ketone bodies derived from triglyceride metabolism [19,20,21]. These ketones are an important energy source for the aging brain as well, as aging can cause mitochondrial dysfunction and glucose hypometabolism [12,22,23,24] which can lead to oxidative stress and free radical damage associated with cognitive impairment [17].

Currently, there is no treatment to cure CCD. Different medications, diets, and dietary supplements (DSs) have been described to ameliorate the clinical signs of CCD. In addition, certain diets and DSs have also been shown to delay the onset of clinical signs in aging dogs [25,26]. The monoamine oxidase B inhibitor L-deprenyl (selegiline) [7,8] increases levels of catecholamines in the cortex and hippocampus and may provide neuroprotection against free radicals [27], improving the clinical signs of CCD.

However, improving all clinical signs of CCD [7] with selegiline is not possible, nor is it effective in all dogs. High variability in results with selegiline has been described and some dogs do not improve their cognitive ability with its supplementation [27,28,29,30,31]. Propentofylline stimulates blood flow to the brain and other organs, such as the heart and skeletal muscles, and may therefore improve dullness, depression, and lethargy of elderly human patients [7]. Limited evidence exists to confirm the benefits of propentofylline on cognitive abilities in people [32] and none in dogs. Further treatment options are needed to manage CCD. Recent studies show that apoaequorin, a calcium-buffering protein, might have neuroprotective properties [33], and may be used to treat CCD or AD. Memantine, which could inhibit ß-amyloid accumulation and prevent neuronal death [34] could also be used to treat CCD, as well as the butylcholinesterase inhibitor “BChEi” [35].

Due to the current lack of effective medication to treat CCD, the management of such patients can be difficult. Therefore, dietary intervention has become an increasingly important part of CCD therapy as the beneficial influence of diet and DS on CCD has been described by various authors [36,37] and is especially effective, if given at an early stage [38]. In regard to the reduced glucose metabolism of the aging brain, supplementation with medium-chain triglycerides (MCTs) can have a positive and long-lasting influence on a dog’s cognition [39,40,41] as these triglycerides are converted readily into the ketone bodies beta-hydroxybutyrate, acetone, and acetoacetate by the liver and the astrocytes in the brain; thus, serving as an alternative energy source for the brain [19,20,21,42].

Due to the brain’s high susceptibility to oxidative damage (caused by mitochondrial dysfunction), supplementation with antioxidants may reduce cognitive impairment [17,43]. S-adenosylmethionine (SAMe), an endogenous methyl donor [44], has been shown to reduce oxidative stress on the brain by increasing glutathione levels and decreasing lipid peroxidation [45,46]. In rats, dogs, and humans, cognitive function improved after supplementation with SAMe [47,48,49,50]. Dietary enrichment with other antioxidants, such as vitamins C and E, L-carnitine, and a-lipoic acid, can also improve memory and learning ability in aged dogs [43] and result in better scores in cognitive testing [37]. B vitamins are vital for normal brain function; deficiencies in B6, 12, and folate, have been described as risk factors for AD [51]. By supplementing B vitamins in combination with antioxidants, omega-3 fatty acids, and arginine, cognitive functions have been shown to improve [26]. The intake of omega 3 fatty acids, docosahexaenoic acid (DHA) and eicosapentaenoic acid (EPA), has been shown to reduce the risk of AD in humans [52], as DHA may reduce the chronic inflammation observed in the pathology of AD and CCD by reducing the availability of arachidonic acid [53]. It may also support brain function and structure by inducing antioxidant defense enzymes; thus, preventing oxidative damage by providing neuroprotection and reducing amyloid accumulation [12,53]. DHA deficiency is one of the risk factors for developing AD [53].

As CCD has a complex, multi-faceted pathophysiology, including several metabolic and pathological changes in the aging brain, nutrient blends containing different DS have been used to target these multiple processes instead of single DS as monotherapy [40,54]. Various studies have examined the benefits of Senilife^®^, which contains phosphatidylserine, Ginkgo biloba extract, vitamin E, and pyridoxine (Vitamin B6), and describe improved CCD symptoms [54,55,56]. Aktivait^®^, a blend of DHA/EPA, vitamin C, N-acetyl cysteine, L-carnitine, a-lipoic acid, vitamin E, acetyl-L-carnitine, CoQ10, phosphatidylserine, and selenium, has also been able to improve the cognitive abilities of aged dogs [57]. Pan et al. tested a brain-protective blend (BPB) containing antioxidants, arginine, B vitamins, and DHA/EPA alone [26] and in combination with MCTs [40] that was able to alleviate CCD symptoms.

Numerous studies have been carried out regarding the nutritional management of CCD. However, because most dogs suffering from CCD are not formally diagnosed by a veterinarian [2,7], it is currently unknown what DS owners actually add to the diet once they start noticing age-related behavioral changes. It also remains unclear which sources owners use to obtain information on dietary supplementation. Different online support groups or non-scientific articles provide recommendations on DS to help manage the clinical signs of CCD. As Berk and colleagues [58] have previously reported in respect of epilepsy management, this information is usually not evidence-based. This study aims to examine how and if dietary interventions for CCD are used by owners. The study investigates the use of DS by patient owners, the sources from which owners obtain their information on supplementation, and what they describe as possible side effects. The results of our study could give veterinarians crucial information when treating a patient with cognitive dysfunction. As different DSs are becoming more popular, the result of this study may help counsel owners with aging dogs before clinical signs occur and enable veterinarians to react more effectively to owners giving unknown or potentially harmful DSs.

## 2. Materials and Methods

A former questionnaire [58] was modified for CCD and optimized with the help of veterinary surgeons, veterinary technicians, and patient owners (Appendix A). From November 2022 to February 2023, a survey was published online via LimeSurvey. Owners of dogs over 8 years of age with age-related behavioral changes were recruited via social media platforms, such as Facebook and Instagram. Participation was open to patient owners regardless of whether a veterinary surgeon had diagnosed their dog with CCD or not. Owners had to give informed consent and agree to privacy policies before they were allowed to enter information into the online questionnaire. Only fully completed questionnaires were considered for analysis.

Data on the patient owner, general patient data (e.g., signalment, pre-existing conditions, medication), the Canine Dementia Score (CADES), a history of a CCD diagnosis by a veterinary surgeon, information on feeding, diets, and the administration of supplements were collected. The participants were then divided into different groups: dogs that had been diagnosed with CCD by a veterinary surgeon and dogs that had not. In addition, owners were categorized by whether they were currently administering supplements, had administered them in the past, or had never administered supplements. Owners were asked to indicate which supplements they administer or administered as mono- or poly-supplementation, for which they could select >25 options and indicate additional supplements in the form of free text. They were asked to indicate reasons they had for or against any giving supplements (captured as free text), how they learned about and purchased supplements, and what were the side effects that they noticed.

### Statistical Analysis

Data were handled using Microsoft Excel and GraphPad Prism (GraphPad Software, Inc., La Jolla, CA, USA). Descriptive statistical analyses were performed. Two-tailed chi-squared tests were used to assess population frequencies.

Data are presented as means with standard deviations.

## 3. Results

Of a total of 1480 received questionnaires, 394 questionnaires were fully completed. Most participants were female (91%) and between 46 and 60 years old (39%). On average, participants were keeping dogs for 24 years (23.54 ± 12.58). The majority of participants were from Germany (86%), followed by the United States (6%), the United Kingdom (2%), Switzerland (2%), and Canada (1%). There were 115 different dog breeds represented, the majority of which were mixed breeds (30%), followed by Labrador Retrievers (7%), Border Collies (3%), Jack Russell Terriers (3%), and Australian Shepherds (3%). Most dogs were female neutered (41%) and were, on average, 12.6 ± 2.4 years old. The mean weight of the dogs was 20.98 ± 12.56 kg.

Two-thirds (67%) of participants reported currently giving their dog nutritional supplements with the aim to improve their behavioral changes, their general health, or other age-related diseases. Almost half of the owners were taking supplements themselves (48%).

### 3.1. Clinical History

The majority of owners stated that their dogs were receiving routine medication such as vaccinations and deworming treatment (46%), medication for brain health (21%), or other medication (43%). Less than one-quarter (24%) of dogs were not on any medication at the time. Two-thirds of the owners reported that their dogs had diseases other than CCD, with the most common being diseases of the musculoskeletal system (44%), such as osteoarthritis or spondylosis. Other common diseases were heart disease (14%), endocrinopathies (13%), and nephropathies (9%). In 31% of dogs, no disease other than CCD was reported.

### 3.2. Cognitive Dysfunction Data

Around a third of dogs (36%) had mild CCD signs according to CADES, 22% showed moderate, 17% showed severe, and 25% had no CCD signs, respectively. Cognitive dysfunction had already been diagnosed by a veterinarian in 69 of 394 dogs, of which 13% showed mild (*n* = 9), 30% moderate (*n* = 21), and 57% (*n* = 39) severe cognitive impairment. The majority of these dogs (81%) were diagnosed by their primary care veterinary surgeon, with only a few diagnosed first by a veterinary specialist—a neurologist (19%).

The minority of dogs were currently receiving medications such as propentofylline (18%), selegiline (2%), or others (1%) with the aim to support brain health. Most owners who were currently, or had in the past, given medication (*n* = 110) reported that it caused no side effects (60%). The most observed side effects were disorientation (30%) and diarrhea (14%).

### 3.3. Nutrition

The majority of dogs were fed dry food (30%) or a mixture of wet and dry food (17%). Raw food was fed in 13.71% of cases and 8.63% were exclusively fed wet food. Most dogs were fed twice daily (64%) and nearly half of all dogs had additional treats several times per day (48%). Food was mainly purchased online (62%). About one-third of owners said they followed the current feeding regime on the advice of their veterinary surgeon (31%), while 17% developed it from information on the internet and 13% from books and magazines. A percentage of 45% indicated “Other” as a source of information. Half of the owners (51%) assessed their dog’s nutritional status with a body condition score of 3/5. Most owners reported spending between one and three hours (67%) per day walking their dog.

Owners whose dog had already been diagnosed with CCD by a veterinary surgeon did not change their dog’s diet (88%), with a few owners reporting a change in their dog’s diet (12%). This was not statistically different from the population that had not been diagnosed by a veterinary surgeon (92%) (*p* = 0.3342). Overall, 8% of dog owners changed their diet when recognizing age-related behavior changes (Figure 1a). There were no statistical differences between the two groups; thus, the following data are mainly presented here for the overall study population. The complete data set stratified into with or without diagnosis of CCD by a veterinary surgeon can be found in the Appendix A, in Appendix A. Feeding prescription diets such as Purina NeuroCare (7%) or Hills b/d (1%) was not frequently reported. Most owners did not feed a therapeutic diet (74%) for age-related behavioral changes or reported feeding a specific diet for a different condition (19%), such as joint or kidney disease. Whether a veterinary surgeon recommended a diet change after the CCD diagnosis is unknown.

### 3.4. Dietary Supplementation

#### 3.4.1. Overall Study Population

Overall, 54% of owners reported giving their dog supplements to treat age-related behavioral changes. A total of 11% of owners gave supplements for this reason in the past and 35% had never used them (Figure 1b).

The most used DSs, overall, were fats and oils (77%), followed by vitamins (48%), amino acids (21%), micronutrients, such as iron, selenium, calcium, and other minerals (17%), and dried herbs and plants (16%) (Figure 2). Few owners reported using combination supplements. A total of 6% of owners used Aktivait^®^, 3% used Denamarin^®^, and 0.4% used Senilife^®^.

Overall, the most used supplements were fish oil (48%) and vitamin B supplements (44%). In addition, many owners reported using DHA or EPA (36%), cannabidiol (CBD) oil (32%), coconut oil (31%), milk thistle (25%), vitamin C (21%), vitamin E (17%), selenium (16%), ginkgo biloba extract (13%), and L-carnitine (13%). A total of 8% of all owners reported using MCT oil in their dog’s diet.

About half (48%) of all owners reported having obtained information about supplement administration from their veterinarian. Other sources indicated were the internet (31%) and other dog owners (18%). Most owners purchased supplements online (76%) or from their veterinary surgeon (20%). Slightly more than a third (40%) had observed side effects associated with DSs in their dog. The most commonly observed side effects were fatigue (21%), increased water intake (20%), and increased urine production (17%). Around half of the owners (54%) would recommend administering DSs, while 41% indicated “I do not know”.

#### 3.4.2. Dogs Diagnosed with CCD

Owners whose dog had been diagnosed with CCD (*n* = 69) often reported giving DSs (71%). The most frequently reported DSs were vitamin B supplements (47%). CBD oil (44%), fish oil (41%), DHA/EPA (35%), and milk thistle (35%) were also frequently given. One-third of these owners reported feeding MCTs (33%).

Out of the group diagnosed with CCD, 41% of owners consulted their veterinarian on the supplementation of their dog. More specifically, 41% of patients with mild, 29% of patients with moderate, and 46% of patients with severe cognitive impairment and an existing CCD diagnosis obtained their supplementation information from their veterinarian.

### 3.5. Non-Use of DSs

About one-third of the participants (35%) reported that they had never given dietary supplements to treat their dog’s age-related behavioral changes. The most common reason (34%) was that it was not known that DSs could be used to treat age-related behavioral changes. Other reasons included “other” (29%), lack of evidence that supplements work (13%), and believing that supplements had no beneficial effect (12%) (Figure 3). Of the people that cited “other” as a reason, 35% stated that their dog did not exhibit clinical signs of CCD and therefore, was not receiving DSs, even though the CADES scale had detected mild to moderate cognitive impairment. Of the dogs that had never received DSs, 30% did not show signs of cognitive impairment.

### 3.6. Cessation of DSs

A total of 11% of participants reported using supplements in the past to treat age-related behavioral changes. Most often, the reason for cessation was the palatability issue of the dog not eating the DSs (22%), the lack of a positive effect (16%), the occurrence of side effects (16%), or “other” reasons (20%) (Figure 4).

## 4. Discussion

The extent to which owners change their dog’s diets or use DSs if the dog is thought to have CCD has not been reported to date. Less than 10% of dog owners change their dog’s base diet when they notice clinical signs of CCD, but around half of them will add DSs to their dog’s diet. There appears to be no difference in owner behavior if the CCD diagnosis was made by a veterinary surgeon or if they noticed the clinical signs themselves. Veterinary surgeons, however, were mentioned as information sources for CCD management by a third of the owners. The others mentioned the internet or literature as a source, while (interestingly) nearly half of the owners did not want to declare where they got the information from. This highlights a need of vets having to take a more active lead in nutritional conversations, as there is a strong link between nutrition and healthy ageing [59].

Around one in ten owners reported having changed their dog’s diet upon diagnosis and seemed to prefer adding DSs to their regular food. Since patients with CCD are typically over eight years old, many of them have pre-existing diseases that are common in geriatric dogs, such as chronic kidney disease [60,61,62], cancer, heart disease, periodontal disease, diabetes, or polyarthritis [63,64], which was reported by many owners in this study as well. As a result of these multiple diseases, patients often have specific dietary requirements [59], and owners may have tried different diets in the past. This can make it difficult to entirely change a dog’s diet to improve its cognitive function. Even though the efficacy of therapeutic diets, such as Purina NeuroCare^®^ and Hill b/d^®^ has been described in studies and clinical trials [25,65,66], most owners in this study stated that they did not feed a therapeutic diet for their dog.

DSs were given, as aforementioned, to around half of the population. Even though numerous studies describe the safety and beneficial effects of DSs on CCD, over one-third of owners reported that they had never given DSs to their dog. It can be difficult for owners to recognize CCD based on the behavioral changes their pet is exhibiting [56]. It is common to dismiss the clinical signs as normal aging and to think of them as untreatable and as “just old age” [3,67]. Some owners in this study stated that they do not use DSs because their dog did not display clinical signs of CCD, even though the CADES score had detected cognitive impairment. These findings concur with Osella et al. [56], who reported that in her study including 124 dogs, 60% of patients showed clinical signs of CDS, yet most owners did not think their dog had cognitive impairment and only 28% of owners sought behavioral consultation.

The most used DS in this study was fish oil, which 47% of dogs consumed, or the derived polyunsaturated fatty acids—docosahexaenoic acid (DHA; C22:6, n-3) and eicosapentaenoic acid (EPA; C20:5; n-3) (36%). These conditionally essential lipids must be obtained through the diet, as de novo synthesis is not possible for mammals [68]. High concentrations of DHA can be found in the brain, and it seems to play a crucial role in neural function [69]. DHA has been shown to improve neural development in young dogs [70]. Several studies have described improved cognitive function in rats and mice [69,71,72], as well as humans [73], and deficiencies in DHA may be associated with cognitive impairment [74]. Furthermore, the intake of DHA reduces the risk of developing AD [52]. EPA has also been shown to improve learning ability in rats [75]. Other health benefits of DHA supplementation in humans [76] and dogs have been described, such as the prevention of heart disease, treatment of dermatologic conditions such as atopic dermatitis, and kidney disease [77,78,79]. Although studies have investigated the benefits of omega-3 fatty acids on dogs’ cognitive abilities when combined with other nutrients such as Phosphatidylserine, vitamins B, C, and E, or other antioxidants, data on the sole use of DHA or EPA in dogs with CCD or old age are lacking [12,80] and further investigation is required.

The second most used DSs in our study were vitamins B, C, and E. It has been described that oxidative damage of the brain may lead to neuronal loss and in turn to cognitive impairment [81,82]. As this damage increases with age [83], a diet enriched with antioxidants, such as vitamins C and E, can slow the progression of cognitive decline and promote healthy aging [17]. Especially when combined with environmental enrichment, antioxidants have been shown to improve the learning ability and memory of old dogs [26,37,43,84]. In human medicine, various studies show that deficiencies in vitamin B may lead to cognitive impairment [51,85] and the supplementation of this vitamin can slow cognitive decline [86]. This is because B vitamins play a crucial role in brain metabolism, as they are important cofactors in numerous methylation reactions in the brain [85,87]. A decrease in vitamin B levels may lead to an inhibition of these reactions, causing a decreased concentration of methionine and SAMe. The methylation of proteins, membrane phospholipids, DNA, and the metabolism of neurotransmitters and melatonin, which is essential for normal neurological functions, is inhibited [85]. Further studies concerning the importance of B vitamins for physiological/neurological functions have described an increase in serum homocysteine (Hcy) due to low concentrations of B vitamins [88], which may be linked to a higher risk for vascular disease [89] and cognitive impairment [89,90] and AD in humans, as Hcy might be neurotoxic and lead to brain atrophy, accumulation of neurofibrillary tangles, and white matter damage (detailed review see [91]). Numerous authors have investigated the beneficial effects of Hcy level lowering with B vitamins on cognitive function in humans [91,92]. However, to this date, no clinical trials have been reported in dogs. When combined with arginine, antioxidants, and omega-3 fatty acids as part of a nutrional blend, B vitamins were able to improve cognition [26], although their specific contribution remains unclear.

Cannabidiol (CBD) was mentioned frequently by owners in this study. CBD is a phytocannabinoid derived from the Cannabis sativa plant [93] and was mostly given in the form of CBD oil. In the last decade, it has been used as an antiseizure drug in rodents [94,95], humans [96], and dogs [97]. The safety and tolerability of CBD have been investigated in healthy dogs, with reported adverse effects being mild gastrointestinal, neurological, or constitutional symptoms and an elevation of serum alkaline phosphatase (ALP) reported by some studies [98,99] (for a detailed review see [100]). It may also be a treatment option for cognitive impairment and behavioral comorbidities of canine epilepsy [101] and for mood or anxiety disorders [102,103]. While the mechanism of CBD’s anticonvulsant effect has been investigated to a certain extent, it is currently unclear how exactly CBD acts as an anxiolytic [103]. The use of CBD for such purposes has not been reported in dogs to date. The use of CBD to treat CCD has also not yet been reported and further research is required in this area.

Different herbs and plants, such as devil’s claw, milk thistle, curcuma, ginger, and brewer’s yeast, were reportedly given by many owners. For most of these herbs, with the exception of milk thistle, there are some studies suggesting potential anti-inflammatory, analgetic, or antioxidant properties in humans; however, a very limited number of studies exist on the safety and efficacy in dogs and there are currently no data regarding the use of any of these plants for the treatment of CCD [104,105,106,107,108,109]. Milk thistle is often used for the treatment of hepatopathy in humans and dogs [110]. Due to its hepatoprotective, antioxidant, cytoprotective, and anti-inflammatory properties, some have suggested that it might prevent liver damage under the chronic administration of antiseizure medication in canine epilepsy. However, as milk thistle can affect the cytochrome P450 activity, and thus changing the pharmacokinetic properties of different antiseizure medications, it may also increase the hepatotoxicity of these drugs [58]. Data on the use of these herbs in dogs with CCD and the effect it could have on commonly used medication are lacking.

Surprisingly, despite the various studies supporting the safety and efficacy of MCT oil to manage CCD, it was not the most commonly used DS. Out of all owners, only 8% reported supplementing their dog’s diet with MCTs. Many owners did, however, mention that they use coconut oil (20%), which is the main source for MCT production. Coconut oil does have a high MCT content but also other longer chain fatty acids. Therefore, the recommendation is to use MCT oils directly [31]. MCT oil was more frequently used in dogs that had previously been diagnosed with CCD by a veterinarian. About a third of these patients were reported to consume MCT oil. The most used oil, however, in this group was not MCT or fish oil but CBD oil, which was given to 44% of dogs, despite there being a paucity of data. This discrepancy between evidence-based treatment and what owners commonly use may be due to a lack of awareness about the existing data. Another reason can be palatability issues with MCT. There is a poorer acceptance of MCT by dogs, especially in doses above 9% of the caloric intake [40].

Contrary to what Berk et al. have reported concerning epilepsy management, around half of owners ask their vet for advice on dietary supplementation. Berk and colleagues [58] reported only one-fifth of the owners having consulted their veterinarian about DS, whereas nearly 40% cited online support groups as their source of information. Owners may be more likely to consult their veterinarians about DS for CCD management because elderly dogs often suffer from several conditions at the same time [111] and owners might fear harming their pet by giving a DS.

This study has multiple limitations, the biggest one being that it is an online questionnaire and thus, based on the sex and age groups of people who answered the questionnaire, this could not be representative for the overall population. This study’s population was biased toward female owners between the ages of 46 and 60. Owners of dogs with severe cognitive impairment might be more likely to consult a veterinarian or gather information elsewhere. Furthermore, as this was an owner-based questionnaire, we did not have clinical records of the patients and did not evaluate them ourselves. Therefore, we cannot exclude including dogs that may have received a false-positive or false-negative diagnosis. However, some of this bias could be considered minimal, as all dogs diagnosed with CDS by a veterinary surgeon also had a positive CADES evaluation for CDS.

The other limitation is that elderly pets have comorbidities; thus, geriatric dogs often receive several DSs for different reasons, such as orthopedic issues such as osteoarthritis. DSs were often reported by owners regardless of why the dog was receiving them, making it unclear whether a dog was receiving supplements as part of the CCD management or for another disease. Geriatric dogs often suffer from multiple diseases at the same time [111] and thus receive various medications and require specific diets. Due to the high individuality of senior patients, a generic, standardized treatment plan is not always applicable [59] and needs to be adapted to each patient’s specific needs. The interactions between DS and certain nutritional requirements of patients because of other diseases need to be considered when discussing or recommending treatment options for CCD.

## 5. Conclusions

Many controlled clinical studies have investigated the efficacy of certain DSs, such as MCTs, antioxidants, or supplement blends as treatment options for CCD. Only around 10% of owners have reported changing the base diet of their dog with clinical signs of CCD, but half of owners in this study have reported using DSs in their dog’s diet, including many DSs that have not yet been investigated regarding their safety or efficacy. This highlights the importance of nutrition in veterinary practice and should encourage veterinarians to discuss this topic with patient owners. Discussing nutrition in general and DS use with patient owners is important, as veterinary surgeons should offer evidence-based recommendations to find the most effective therapy for dogs with CCD and to counsel owners with aging dogs before clinical signs occur.

## Figures and Tables

**Figure 1 animals-13-03056-f001:**
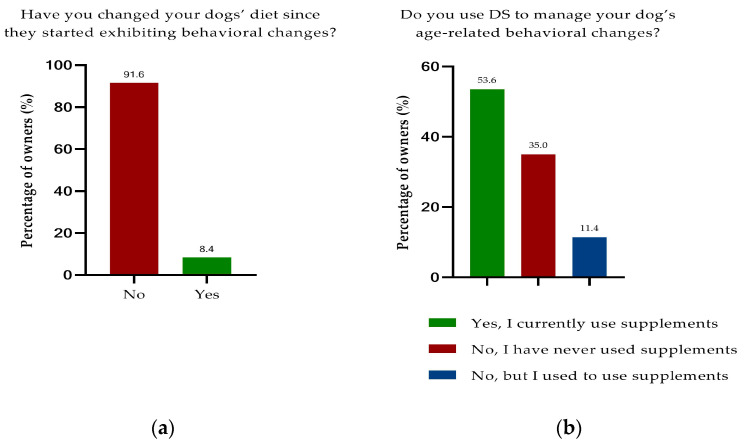
Diet change and the use of dietary supplements to manage age-related behavioral changes. (**a**) Less than 10% of owners changed their dog’s diet to manage behavioral changes. (**b**) About half the owners currently give dietary supplements.

**Figure 2 animals-13-03056-f002:**
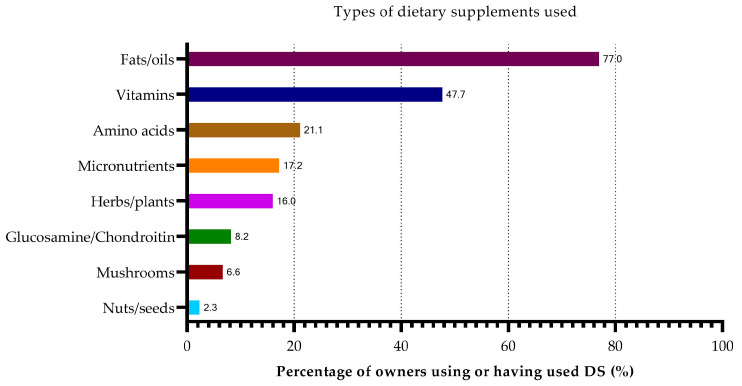
Types of DSs used by owners to manage age-related behavioral changes in percent (%). Around half the owners administer DSs and three out of four dogs receive fats or oils, such as fish oil, CBD oil, and coconut oil in addition to their normal diet.

**Figure 3 animals-13-03056-f003:**
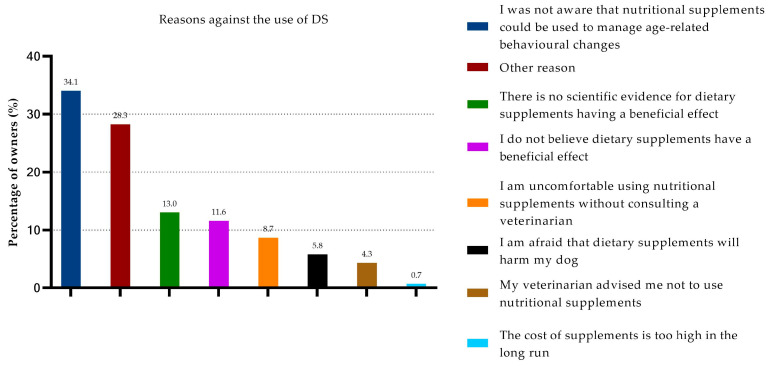
Owners’ reasons not to administer DSs to manage age-related behavioral changes.

**Figure 4 animals-13-03056-f004:**
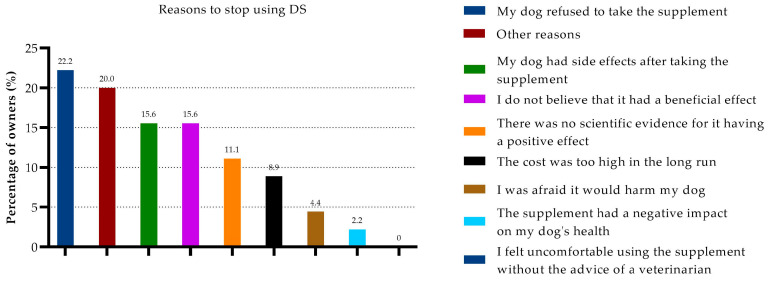
Owners’ reasons to stop administering DSs to manage age-related behavioral changes.

## Data Availability

The data presented in this study are available upon request from the corresponding author.

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
