# Peer review of "Investigating Owner Use of Dietary Supplements in Dogs with Canine Cognitive Dysfunction"

_animals, 2023, doi:10.3390/ani13193056_

Round 1

Reviewer 1 Report

This is a good review of neurochemistry of CCD and it is an excellent flag of what is still left unaddressed by veterinarians (and perhaps, untaught to veterinary students???).  I made a number of comments to clarify the text on the ms. and asked some questions about linking extent of fn to whether a vet was involved in the dx. 

Reviewer 2 Report

The article provide insights in the application of dietary supplements on dogs with canine cognitive dysfunction. The subject matter is of interest to readers of Animals. The general design is appropriate for a research article, but it do not looks like the final version, which was submitted.

Specific comments:

The article was largely fine, although the statistical analysis is poor, no specific data analysis such as logistic regression was used. Why only fully completed questionnaires were used for analysis?

Dog handling such as go for a walk with the dog are different among dog owners from different countries, so why dog owners from different countries and continents were mixed in the data set?

Line 305: one in ten people ?
